# Crack Resistance of Prestressed Reinforced Concrete Beams with Wire Rope Reinforcement

**DOI:** 10.3390/ma16196359

**Published:** 2023-09-22

**Authors:** Saidgani Yusufkhojaev, Rakhimbay Yusupov, Xikmat Alimov, Jaloliddin Makhmudov, Eunsoo Choi

**Affiliations:** 1Department of Building Construction, Tashkent University of Architecture and Civil Engineering, Tashkent 100114, Uzbekistan; saidganixodja@gmail.com (S.Y.); raximbay1951@gmail.com (R.Y.); xikmat.phd88@gmail.com (X.A.); makhmudov9910@gmail.com (J.M.); 2Department of Civil and Environmental Engineering, Hongik University, Seoul 04066, Republic of Korea

**Keywords:** heavyweight reinforced concrete, ceramporite concrete, wire rope reinforcement

## Abstract

One of the main objectives of capital construction worldwide is to increase the efficiency of construction. To achieve this, a significant improvement in the quality of concrete and reinforced concrete is required. Nowadays, in developed countries, one of the key directions for enhancing the efficiency of concrete and reinforced concrete is reducing their density by using inexpensive and high-quality local porous fillers. In recent years, the use of concrete structures with porous fillers based on ceramporite in load-bearing reinforced concrete structures has allowed for the improvement of their technical and economic indicators and the reduction in their own weight by up to 35%. This, in turn, is considered an essential factor in seismic zones, and therefore, special attention is paid to these issues. This article presents the data of experimental and theoretical studies obtained from investigating the crack resistance of prestressed lightweight reinforced concrete flexural elements with wire rope reinforcement. The main factors influencing crack resistance were considered, including the effects of two types of porous filler (ceramporite), two types of concrete, concrete strength, percentage of reinforcement, and level of precompression. The tests were conducted using short-term and repeated loadings. Based on the analysis of the obtained data, some refinements have been made to the methodology for calculating the crack resistance of such structures according to current design standards.

## 1. Introduction

In recent years, the use of concrete structures with porous fillers based on ceramporite (ceramporite is a new artificial porous filler obtained by firing a mixture of clay and kaolinite, including bentonite) in load-bearing reinforced concrete constructions has allowed for the improvement of their technical and economic indicators and the reduction in their weight by up to 35%. This reduction in weight is considered an important factor in seismic zones, and as a result, special attention is given to these aspects.

Bendable elements, which can be flexed until they break, made of cable-reinforced ceramporite concrete, have not been studied in detail up to the present time. This circumstance complicates the comprehensive consideration of their properties during the calculation and design of load-bearing reinforced concrete structures. This primarily applies to the peculiarities of the prestressing effect, resistance of prestressed elements to repeated loads, conditions of crack opening and closing, and other aspects.

Research on the stress–strain behavior of bendable prestressed elements has mainly been conducted using various types of lightweight concrete and bar reinforcement. However, specific and targeted studies on the peculiarities of the behavior of bendable prestressed elements made of ceramporite concrete with cable reinforcement have scarcely been conducted.

In order to facilitate the wide application of these materials, it is necessary to carry out new experimental and theoretical research to refine and improve the calculation and design requirements of regulatory documents for the design of such prestressed reinforced concrete structures.

Despite the availability of experimental research on the crack resistance of prestressed reinforced concrete structures, we believe that their calculation remains insufficiently addressed in current design standards. This is particularly relevant when considering the application of new types of lightweight concrete in combination with wire rope reinforcement.

It is worth noting that the provisions of (construction norms and regulations) KMK 2.03.01-96 [1] concerning crack formation calculations do not consider the influence of precompression on changes in concrete tensile strength. Additionally, the calculation of crack opening does not account for the behavior of reinforcement beyond the elastic range or the variations in mechanical properties of prestressed reinforcement. Furthermore, the current norms lack guidelines for considering an increase in a_crc_ during scenarios involving multiple static loadings.

As is known, the existing norms are primarily based on experimental research conducted on reinforced concrete structures using heavy concrete [2,3,4]. The technical advantages of employing high-strength wire rope reinforcement in ceramporite concrete flexural elements, as well as the effects of prestressing, have not been sufficiently assessed within these norms.

During the process of crack formation in reinforced concrete structures, three stages are commonly distinguished: the initiation of cracks, when they may still be invisible; the appearance of cracks, when they become detectable to the naked eye; and crack propagation. According to this theory, the moment of crack appearance coincides with the moment of their initiation. The opinion that the emergence of visible cracks is preceded by an intensive process of microcrack formation has been repeatedly expressed by various researchers [5,6,7,8,9]. The calculation of crack width is based on a methodology developed by Prof. V.I. Murashev and further augmented for prestressed structures in subsequent years by himself and other researchers [10,11,12,13,14,15,16]. The influence of the protective layer on crack width has been studied both in the CIS countries [17,18,19,20]. In the work of Sh.A. Khakimov and S.A. Dmitriev, the effects of the thickness of the protective concrete layer on the bending elements’ crack width were investigated. Based on the recent research results [21], the failure of prestressed hollow-core slabs manufactured by the extrusion method was modeled. In other studies [22,23], investigations have revealed aspects of the behavior of prestressed building slabs, considering the changes in their properties and connections.

Reducing the self-weight of reinforced concrete structures through the use of various lightweight concretes, including prestressed elements based on them, is effective in seismic regions of Uzbekistan. In this regard, research in this direction will contribute to solving a number of issues related to the application of efficient prestressed elements [24,25].

Well-founded results were also obtained in another study [26], which demonstrates that there is a difference in the behavior of prestressed beams reinforced with steel periodic profile and wire rope reinforcement in terms of bending moments corresponding to the formation of the first diagonal crack. Furthermore, in beams with wire rope reinforcement, there is a sudden crack opening accompanied by shear strength loss, unlike beams with bar reinforcement, where crack opening occurs gradually, i.e., a stepwise progressive crack opening is observed. According to the authors, this is attributed to a certain weak adhesion between the wire rope reinforcement and concrete.

In the work [27], aimed at improving the seismic behavior of a damaged reinforced concrete building, based on an analysis of extensive experimental data, it is proposed to install an external steel frame. This led to an increase in its stability and reliability. This confirms the effectiveness of the proposed solution for strengthening reinforced concrete structures, which is also carried out with the use of prestressed reinforcement.

The present study is devoted to the experimental and theoretical investigation of the behavior of bendable ceramporite-reinforced concrete elements with cable reinforcement under short-term and repeated loading, followed by the development of suggestions to improve the calculation methodology for such structures.

## 2. Materials and Methods

To address the stated objectives, 36 experimental reinforced concrete beams were fabricated. These beams had a rectangular cross-section measuring 12 cm by 24 cm and a length of 320 cm. Two types of lightweight concrete were used in the experimental study: ceramporite concrete, which is a lightweight concrete based on ceramporite (a synthetic porous filler made from local raw materials), and ordinary heavyweight concrete. As for the longitudinal working reinforcement, wire rope reinforcement of K-7 class with diameters of 9 mm and 12 mm was utilized.

To manufacture experimental beams made of ceramporite concrete, two types of ceramporite obtained and local raw materials were used. The optimal compositions for the granulation of ceramporite filler are designated as follows (in %): loess—75; kaolinite—15; coal—10. The primary minerals of the raw components are SiO_2_ (55–58%) and Al_2_O_3_ (12–25%).

The ceramporite filler is produced using the following technology: loess-like clay is sieved through a 2.5 mm sieve, thoroughly mixed with kaolinite and coal that have been sieved through a 2.5 mm sieve, and then moistened to achieve a moisture content of 20–30% by weight. Granules are formed using a disc granulator. The ceramporite was sintered at 1250 °C. The obtained granules are dried to an air-dry state and then directed to be fired in a rotary kiln. The main characteristics of ceramporite porous fillers are presented in Table 1.

As a fine filler, quartz sand from the Chinaz quarry with a bulk density of 1450 kg/m^3^ and a fineness modulus of 2.2 was used. As a binder, Portland cement from the “Ahangaran Cement Plant” with a strength of 39.5 MPa was applied. The specific surface area of the cement is 3500 cm^2^ per gram. The normal density of the cement paste is 27%. According to GOST 310.3-76 “Cements. Methods for Determination of Normal Density, Setting Times, and Uniformity of Volume Change” [28], the normal density of cement paste is characterized by the amount of mixing water, expressed as a percentage of the cement mass. The compositions of ceramporite concrete per cubic meter of concrete are provided in Table 2.

The concrete was prepared using a forced-action concrete mixer and the molds for the twin beams, as well as the corresponding number of control cubes, were filled with a single batch. The poured concrete in the molds was compacted using a deep vibrator. All the experimental beams and cubes were stored under natural conditions after concreting. The transfer of forces from the support to the concrete was carried out at cube strengths of 22–26 MPa and 16–20 MPa, respectively, for lightweight and heavyweight concretes.

To conduct the experimental investigations, a testing program was developed, according to which the reinforced concrete beams were divided into three groups based on the types of concrete, reinforcement, and precompression level. The experimental beams were subjected to external static loads through two concentrated forces positioned at a distance of 110 cm from the support on both sides.

The force was applied using hydraulic jacks DG-100 (Moscow, Russia) (see Figure 1), connected to the pumping station ISR-400 (Moscow, Russia). Deformations of the concrete and reinforcement were measured using strain gauges, load cells, and dial indicators (0.01 mm).

For the production of experimental specimens, rigid steel molds were used. Two strands were tensioned on each power form in sequence, and on each strand, two twin specimens were fabricated. The tensioning of the reinforcement was carried out using a hydraulic jack DP-30-200 (Moscow, Russia) with a manual pumping station drive. The magnitude of the force in the reinforcement during tensioning was monitored using a pressure gauge and by measuring the deformations of the reinforcement.

Figure 1 and Figure 2 show, respectively, the overall view of a rigid steel loading frame used for testing prestressed beams and the observation of the beams prior to conducting the tests. A total of 12 experimental reinforced concrete beams were fabricated for each type of concrete. Each set of beams consisted of 6 beams with longitudinal working reinforcement of 2Ø9 K-7 and 6 beams with reinforcement of 2Ø12 K-7. Among these 6 beams, each set comprised three pairs of twin beams. The first pair had a high level of precompression, the second pair had a moderate level of precompression, and the third pair had no precompression.

Experimental values of the characteristics of the used reinforcement are provided in Table 3.

In the experimental beams with longitudinal working reinforcement of 2Ø9 K-7, wire reinforcement of Vr-II with a diameter of 5 mm was installed in the compressed zone to prevent crack formation during the release of precompression. In the beams with longitudinal working reinforcement of 2Ø12 K-7, a single wire rope reinforcement with a diameter of 9 mm was installed in the compressed zone for the same purpose. Transverse reinforcement was provided by using wire loop ties of Ø5 Vr-1, placed behind the pure bending zone at a spacing of 10 cm. Figure 3: The reinforcement scheme of the experimental beams is presented in Figure 1.

The experimental moment of crack formation was determined when testing beams visually and using a microscope with a 24-fold increase, and then refined by analyzing the graphs of concrete deformation in the compressed and tension zones, tension reinforcement, and deflections.

The strength of ceramporite concrete and heavyweight concrete was determined by testing cubic concrete specimens with 15 cm side length, manufactured from the same concrete mix used for the experimental beams. As a result of the tests, the strength of ceramporite concrete on the day of testing the experimental beams was in the range of 31–33.5 MPa, while for heavyweight concrete, it was in the range of 26–29 MPa.

The results of the experimental studies conducted on samples of reinforced concrete columns to determine concrete strength using non-destructive methods were compared with results obtained from the cubic specimens [29]. In these studies, both the experimental columns and cubic specimens were made from the same composition. A detailed analysis of the obtained results enabled the estimation of the real strength of the concrete and the formulation of predictive equations. This leads to the conclusion that in experimental investigations of reinforced concrete structures, it is advisable to use non-destructive methods alongside destructive methods to assess the actual strength of the concrete structures.

## 3. Results and Discussion

Figure 4 illustrates the deformations of strain gauges attached to the lower surface of the beam, which are most sensitive to crack formation. As seen in the figure, long before the formation of cracks, the deformations occur uniformly. As the load increases, the uniformity is disrupted due to the plastic deformation of the concrete, and the readings of the strain gauge located on the surface of future cracks increase more intensively. At this stage, visible cracks had not yet been detected.

In our research, the experimental value of the crack formation moment was determined as an intermediate value between the moment of visible crack formation and the moment at which a significant increase in deformation occurs, as measured by strain gauges attached to the tensioned surface of the beams.

The first transverse cracks in the zone of pure bending appeared at different relative loading levels, depending on the percentage of reinforcement and the degree of prestressing.

In beams without prestressing and a reinforcement percentage of 0.4%, cracks formed in the zone of pure bending under a load of 0.10–0.12 of the experimental destructive test, and for beams with μ = 0.7% under a load (0.07–0.08) $M_{u}^{exp}$.

This phenomenon can be explained by the fact that with an increase in the percentage of reinforcement, the breaking load increases to a greater extent than the load at which cracks appear. It should be noted that in beams (without prestressing) both from ceramporite concrete and ordinary heavy concrete, immediately after the appearance of cracks, they develop by 1/2 and 2/3 of the section height. A further increase in the load leads to a slower development of cracks along the height and their opening, the appearance of new cracks in the zone of pure bending and beyond. The cracks that appeared in these sections were initially oriented normally to the axis of the beams, and then, as the load increased, they deviated more and more towards the place of application of the load. The process of their formation and development basically ends at a load of 70–80% of the destructive load.

It should also be noted that in beams with a percentage of reinforcement μ = 0.7%, the development of cracks occurs gradually and, before destruction, their height is somewhat lower and the opening width is much less than in beams μ = 0.4%.

At the time of formation and development of cracks, the stress in the tensile reinforcement is significantly affected: with its increase, later formation of cracks occurs, and their propagation along the height and towards the supports slows down. In prestressed beams with a reinforcement percentage of 0.4%, the first cracks were formed at a moment of (0.33–0.48) $M_{u}^{exp}$, which reached (0.58–0.64) h upon destruction along the height. In prestressed beams with μ = 0.7%, the formation of cracks was recorded at (0.41–0.47) $M_{u}^{exp}$, which reached (0.46–0.50) h upon destruction in height (Figure 5).

The theoretical moments of crack formation were determined in accordance with KMK 2.01.03-21 [1], and also taking into account the effect of pre-compression on the change in the mechanical properties of the concrete, according to the formula:(1)$$M_{crc}=K_{bt.ser}\cdot R_{bt.ser}\cdot W_{pl}\pm P(e_{op}+r)\;\;\;\;\;(KN\cdot sm)$$where P is the pre-compression force at the time of testing the beams, taking into account losses;

$K_{bt.ser}$ is the coefficient taking into account the decrease in tensile strength of concrete caused by long-term pre-compression of concrete;

r is the distance from the center of gravity of the reduced section to the core point most distant from the tensile zone, the cracking of which is being checked.

The value of *r* for bending prestressed elements is determined by the formula:(2)$$r\;=\varphi {\cdot W}_{red}\cdot A_{red}\;\;\;(sm)$$where:(3)$$\varphi \;=1.6-\sigma_{b}\cdot R_{b\cdot ser}$$

For specimens with non-stressed reinforcement, the value $M_{src}$ was determined taking into account the negative influence of shrinkage deformations of ceramporite concrete, according to the formula:(4)$$M_{src}=R_{bt\cdot ser}{\cdot W}_{pl}{-P}_{se}(e_{op}+r)\;\;\;\;\;\;\;(KN\cdot sm)$$
(5)$$P_{se}=\sigma_{s.se}(A_{s}+{A^{\prime }}_{s})\;\;\;\;\;\;\;(KN)$$where $\sigma_{s.se}$ is the stresses in the reinforcement caused by shrinkage.

When comparing the actual moments of cracking with the theoretical ones, calculated according to KMK 2.03.01-21, it turned out that for the prevailing number of samples, it was characteristic that the theoretical moments exceeded the $M_{src}$ experimental values $M_{src}^{exp}$, which in some cases reached 10.73%.
$${\sum ∆}_{1}^{2}=972.8;\;{\sum ∆}_{2}^{2}=402.4$$

Note: The beam codes L and T indicate that they are made of lightweight and heavyweight concrete, respectively. The prestressed reinforcement class is specified.

As can be seen from Table 4, the introduction of the coefficient $K_{bt\cdot ser}$ into Formula (1), which takes into account changes in the tensile strength of concrete, leads to an improvement in the convergence of the theoretical and experimental moments of cracking.

An analysis of the experimental data shows that in beams without prestressing, after the appearance of cracks, they immediately cross the tensile reinforcement and develop upward by (0.46–0.64) h of the beam height.

In beams made of ceramporite concrete and heavy concrete at a loading level of 0.6 M_u_, 11–12 and 8–9 cracks, respectively, were observed in the “pure” bending zone, which were located in increments of 6–9 cm and 8–12 cm, respectively.

In prestressed beams at a load level of 0.6·M_u_, 8–10 and 7–9 cracks were observed in the “pure” bending zone for ceramporite concrete and heavyweight concrete, respectively. In beams made of ceramporite concrete reinforced with 0.4% and 0.7% reinforcement, the cracks were spaced at intervals of 7–11 cm and 8–13 cm, respectively. In beams made of heavyweight concrete, the distance between cracks was approximately 8–12 cm and 9–14 cm, respectively.

In our experiments, the type and strength of concrete had practically no effect on the crack opening width.

As the experimental data show, the determining factors for the width of the crack opening are the level of compression, the percentage of reinforcement and the stress in the tensile reinforcement.

Figure 6 and Figure 7 show the results of measurements of the crack opening width depending on the stresses (in prestressed beams—the increment of stresses from the action of an external load) in the reinforcement of the tension zone.

As can be seen from these figures, up to the stress in the tensile reinforcement equal to the conditional yield strength, the relationship between the stress in the reinforcement and the crack opening width can be assumed to be linear with great certainty.

The ratio of the maximum crack width to the average ${ɑ}_{crc}^{exp}$/${ɑ}_{m.crc}^{exp}$ with a decrease in the percentage of reinforcement of beams, it noticeably decreases.

Attitude ${ɑ}_{crc}^{exp}$/${ɑ}_{m.crc}^{exp}$ also decreases with increasing load level. So in beams, with µ = 0.4% ${ɑ}_{crc}^{exp}$/${ɑ}_{m.crc}^{exp}$ = (1.1–1.27), and in beams with µ = 0.7%, this ratio is (1.2–1.44) at M = 0.6 M_u_, and at a loading level of 0.8 M_u_, this ratio is equal to (1.1–1.2) and (1.13–1.27), respectively.

Prestressing has a significant effect on the width of the opening of normal cracks in ceramporite-reinforced concrete beams. Thus, an increase in the level of compression σ_bp_/R_bp_ from 0 to 0.72 in beams with a percentage of reinforcement μ = 0.4% at the loading level M = 0.6 M_u_ reduces the average crack width from 0.39 mm to 0.033 mm, and in beams with µ = 0.7% this value is 0.26 mm to 0.025 mm.

The theoretical values of the maximum crack opening width were determined by the KMK 2.03.01-21 formula with the actual characteristics of the materials.

At the same time, satisfactory convergence between theoretical and experimental values was achieved. The significant deviation of experimental crack width values from theoretical values at loads close to failure can be explained by the fact that in beams with µ = 0.4%, the reinforcement operates with large plastic deformations (the stress in it exceeds the yield limit), which leads to an increase in the experimental crack width.

During the determination of $a_{crc}$ according to the recommendations of the norms, only the elastic behavior of the reinforcement with a constant value of the modulus of elasticity of steel is considered. Additionally, the norms do not take into account the changes in the mechanical properties of the wire rope reinforcement caused by the effect of prestressing.

To improve the convergence of experimental and theoretical values, it is proposed to incorporate the variable modulus of elasticity of the wire rope reinforcement, E’_s_, into the formula of the design code 2.03.01-21.

Theoretical values of $a_{crc}$, calculated taking into account the variable modulus of elasticity and the influence of prestressing on the mechanical properties of the reinforcement, closely match the experimental values, even at high stresses in the reinforcement.

According to the experimental program, one beam from the twin specimens was subjected to multiple repeated loadings up to the moment value where the crack opening width, $a_{crc}$, reached 0.15 mm. Subsequently, the beam was gradually unloaded and subjected to repeated loadings. In the pre-stressed specimens, an increase in the crack opening width was observed only after the third loading cycle. The fourth and fifth loadings did not result in a significant increase in the crack opening width, indicating that the fifth loading was the final one.

In accordance with the recommendations of [2,3,4], the load at which the crack opening width decreased to 0.03 mm was taken as the moment of crack closure in our studies, bearing in mind that such a state does not pose a danger to reinforcement corrosion.

As experimental data have shown, that with an increase in the prestress value from 0 to 0.9 σ_0.2_, the ratio of the moment at which acrc\u003d 0.15 mm to the breaking one increases almost linearly in elements of both heavy and ceramporite concrete. In the latter, the level of moments corresponding to the formation of cracks acrc = 0.15 mm is lower than in elements made of heavy concrete by 6%−12%, depending on the magnitude of the prestress (Figure 8).

It should be noted that at the same level of the bending moment, the crack opening width during unloading in all cases exceeds the value of $a_{crc}$ measured at the same level during loading. This can be explained by the displacement of the sections in the crack itself, as well as the obstacle to the restoration of reinforcement deformations due to inelastic deformations of concrete in the tension zone. Although, in all beams tension reinforcement worked in the elastic stage.

In all non-stressed specimens, the upper level of loading was 2–3 times less than in the prestressed specimens. Reloading resulted in an increase in the width of the crack opening compared to the previous loading, but this increase in the width of the crack opening was slightly larger compared to prestressed ones. 

The crack openings during the load reduction in prestressed beams varied depending on the reinforcement. After the complete removal of the load, beams with a reinforcement percentage of 0.4% exhibited residual crack widths of 0.04–0.05 mm, while beams with µ = 0.7% had slightly smaller widths (0.02–0.04 mm). This can be explained by the fact that elements with lower reinforcement percentages are more sensitive to overloads.

In all prestressed beams, crack closure was observed, although some of them experienced losses in the prestressing of the reinforcement due to its relaxation.

Stresses in concrete at the level of tensile reinforcement ($\sigma_{bp}$) during crack closure were determined by the following formula:(6)$$\sigma_{bp}=\frac{P\left(e_{op}+r\right)-M_{3}}{W_{red}}\;\;\;\;\;\;\;(MPA)$$

*P*—prestressing force at the time of testing beams, accounting for losses; *e_op_*—eccentricity of the prestressing force *P* relative to the centroid of the transformed section; *r*—distance from the centroid of the transformed section to the core point furthest from the tensioned zone; *M*_3_—cracking moment; *W_red_*—resistance moment of the transformed section of the element for the extreme tensioned fiber.

An analysis of the experimental results (Table 5) shows that the ratio of the crack opening width under repeated loading to the crack opening width at the first loading φ_l_ depends on the level of external load, the degree of prestressing and other factors. The ratio φ_l_ = ${ɑ}_{crc5}^{exp}$/${ɑ}_{m.crc1}^{exp}\;$in our experiments fluctuated within 1.07–1.2. On average, it can be assumed to be equal to 1.15.

## 4. Conclusions

The formation and development of cracks in bendable elements made of ceramporite concrete with cable reinforcement are similar to those in elements made of heavy concrete;There is a linear relationship between the level of crack formation and the level of compression of prestressed ceramporite concrete bendable elements;Accounting for the influence of prolonged prestressing of ceramporite concrete on its mechanical properties ensures the best convergence of theoretical and experimental cracking moments when using the coefficient $K_{Rbt.ser}$ in the design formulas of the codes;When calculating the crack width of ceramporite-reinforced concrete elements under short-term loading, the expression provided by the codes can be used by substituting the variable modulus of elasticity of the reinforcement $E_{s}^{\prime }$;In bendable elements with cable reinforcement, both in heavy concrete and ceramporite concrete, as the level of prestress increases from 0 to 0.9 σ0.2, the ratio of the moment at which a_crc_ = 0.15 mm to the ultimate moment increases almost linearly;In ceramporite concrete bendable elements with cable reinforcement, the level of moments corresponding to crack opening a_crc_ = 0.15 mm is lower by 6–12% compared to elements made of heavy concrete, and it depends on the magnitude of prestress;The crack width under repeated loading exceeds the crack width under initial loading, which can be determined using the formula provided by the design codes with the coefficient φ_b_ equal to 1.15;The experimental research data confirm the reliability of the code requirements regarding crack closure in bendable elements of the 2nd category of crack resistance made of ceramporite concrete with cable reinforcement.


**Practical results of the research:**
New experimental results on the crack resistance of prestressed beams made of ceramporite concrete reinforced with wire rope reinforcement were obtained based on the conducted research;An improved calculation method for flexural elements based on the characteristics of ceramporite concrete and the real deformation diagram, considering the effects of wire rope prestressing, was proposed;Formulas for determining the mechanical properties of concrete and reinforcement, the degree of prestressing of the reinforcement, and the stiffness of crack-free sections taking into account concrete compression were introduced;Formulas for determining the crack resistance of prestressed ceramporite concrete beams reinforced with wire rope reinforcement subjected to short-term and repeated loads were recommended.


In our opinion, further experimental and theoretical research on the stress–strain behavior of prestressed flexural elements made of ceramporite concrete with wire rope reinforcement should focus on studying their performance under sustained loading, particularly considering the influence of adverse environmental conditions.

## Figures and Tables

**Figure 1 materials-16-06359-f001:**
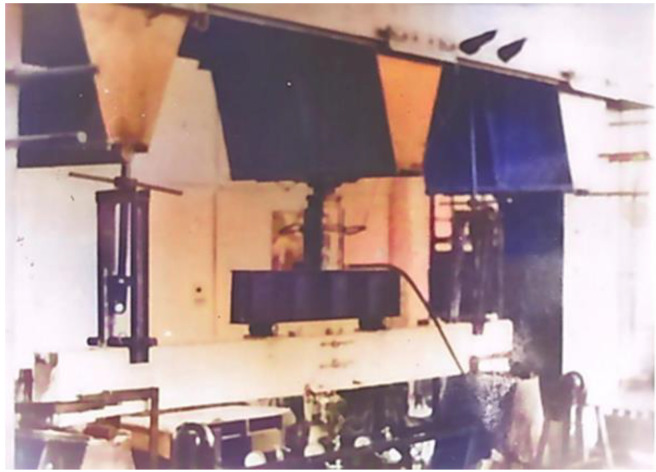
Hydraulic jack DG-100.

**Figure 2 materials-16-06359-f002:**
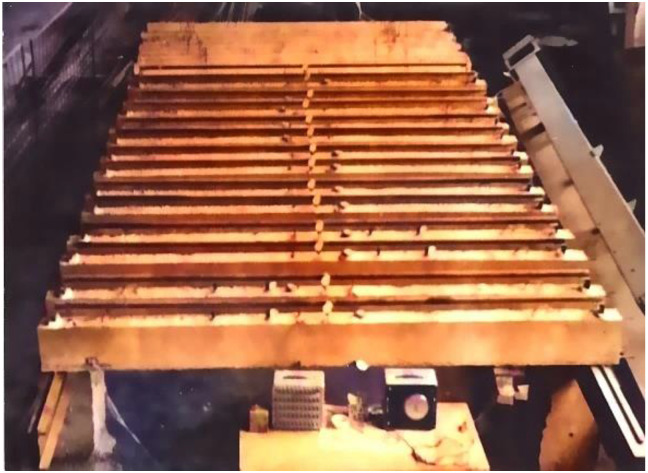
Observation of the beams.

**Figure 3 materials-16-06359-f003:**
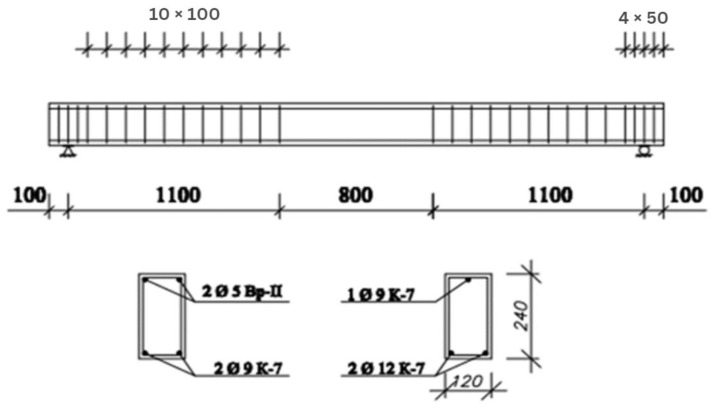
The reinforcement scheme of the experimental beams. Unit: mm.

**Figure 4 materials-16-06359-f004:**
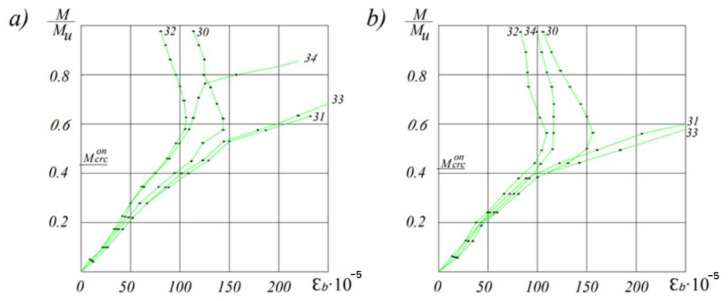
Deformations of the tensioned surface of the beams depending on the load level for the ceramporite concrete reinforced beams of the first (**a**) and second (**b**) types.

**Figure 5 materials-16-06359-f005:**
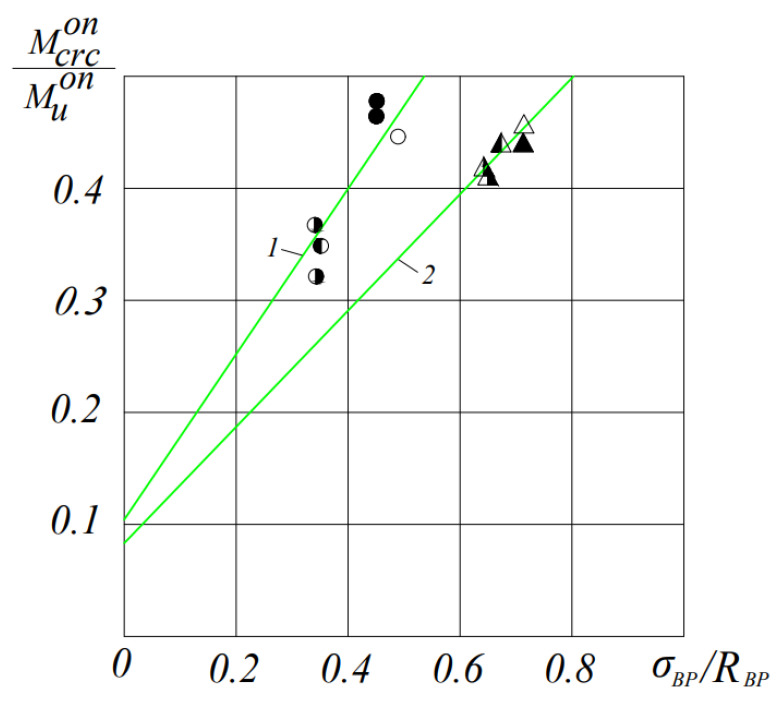
The dependence of the moment of cracking in experimental beams made of ceramporite concrete on the level of compression: 1—experimental beams with μ = 0.4%, 2—experimental beams with μ = 0.7%. ⚫-BL19V-1, ○-BLP9V-1,2, ◐-BL19C-1,2, ◑-BLP9C-1,2, △-BLP12B-1,2, ▲-BL112B-1,2, ◭-BL112C-1,2 and ◮-BLP12C-1,2 (see Table 4 also).

**Figure 6 materials-16-06359-f006:**
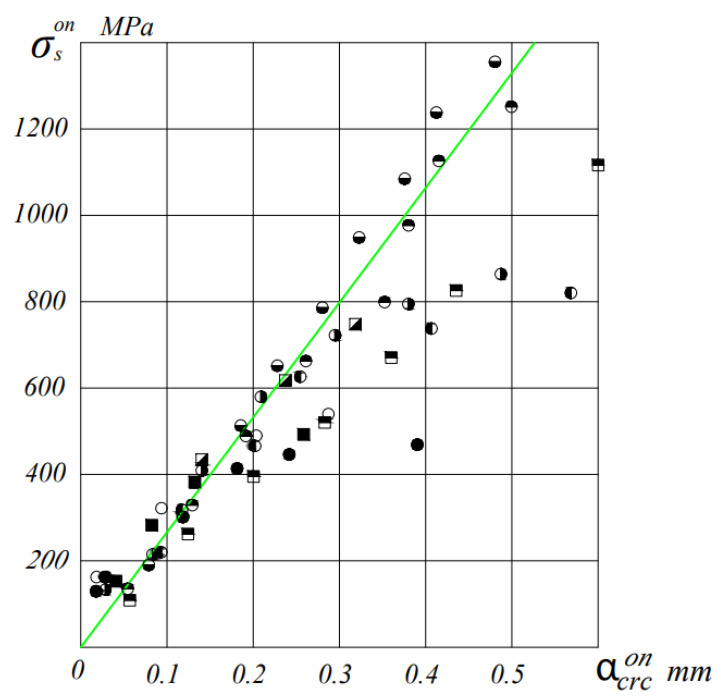
Average crack width opening in relation to the increment of stresses in the reinforcement $\sigma_{s}^{on}$ for beams with µ = 0.4%. I—value $a_{crc}$ according to KMK 2.03.01-21. ⚫-BL19V-1, ○-BLP9V-1,2, ◐-BL19C-1,2, ◑-BLP9C-1,2, ◓-BL19O-1,2, ◒-BLP9O-1,2, ■-BT9V-1,2, ◪-BT9C-1,2 and ⬒-BT9O-1,2.

**Figure 7 materials-16-06359-f007:**
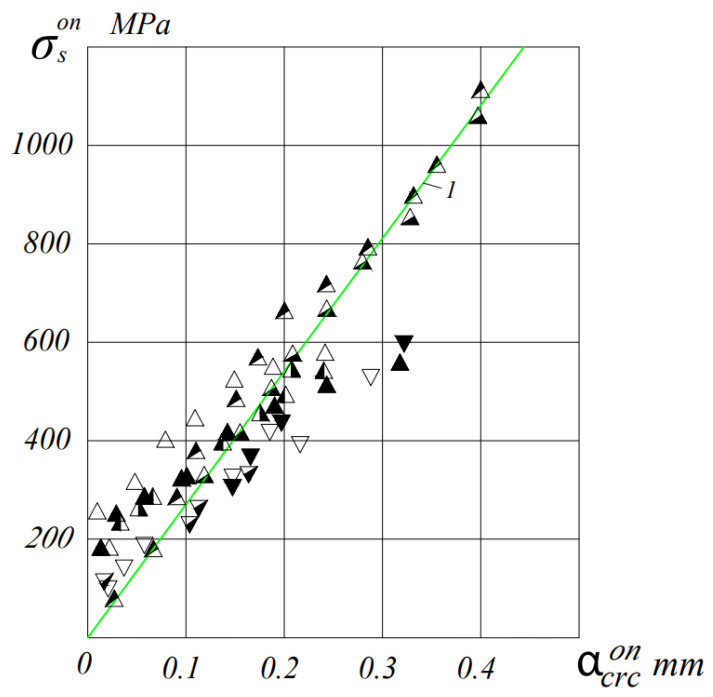
Average crack width opening in relation to the increment of stresses in the reinforcement $\sigma_{s}^{on}$ for beams with µ = 0.7%. I—value $a_{crc}$ according to KMK 2.03.01-21. △-BLP12B-1,2, ▲-BL112B-1,2, ◭-BL112C-1,2, ◮-BLP12C-1,2, ⧩-BT12C-1,2, ▽-BT12B-1,2 and ⧨-BLP12O-1,2.

**Figure 8 materials-16-06359-f008:**
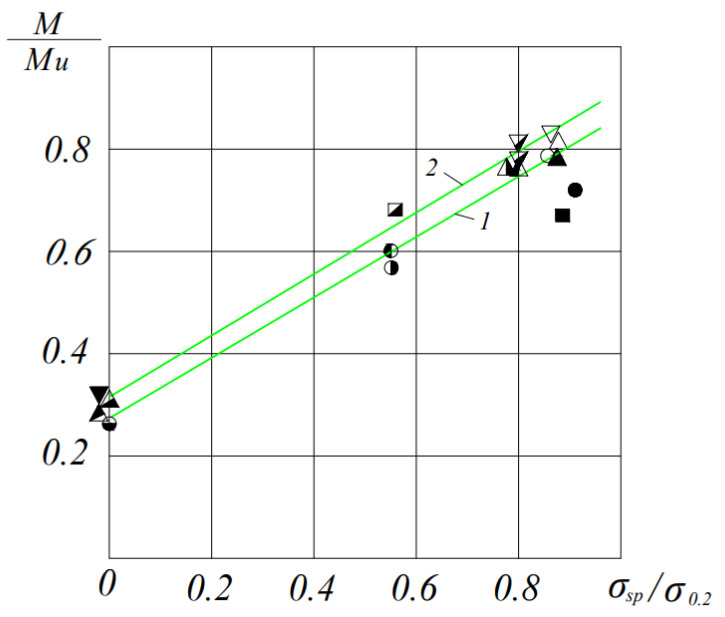
Dependence of the ratio M/M_u_ on the stress σ_sp_/σ_0.2_ for a crack opening width of $a_{\mathrm{crc}}$ = 0.15 mm: 1—beams of ceramporite concrete; 2—heavy concrete beams. ⚫-BL19V-1, ○-BLP9V-1,2, ◐-BL19C-1,2, ◑-BLP9C-1,2, ◒-BLP9O-1,2, ■-BT9V-1,2, ◪-BT9C-1,2, △-BLP12B-1,2, ▲-BL112B-1,2, ◭-BL112C-1,2, ◮-BLP12C-1,2, ⧩-BT12C-1,2, ▽-BT12B-1,2 and ⧨-BLP12O-1,2.

**Table 1 materials-16-06359-t001:** Main Physicomechanical Properties of the Used Ceramporite Fillers.

Indicators	Unit of Measurement	Ceramporite Type 1		Ceramporite Type 2	
		Particle Sizes, mm			
		5–10	10–20	5–10	10–20
Bulk density	Kg/m^3^	850	830	810	800
Intergranular void volume	%	44.3	45.8	45.1	46.3
Porosity	%	48.2	49.1	49.3	49.8
Compressive strength in a cylinder	MPa	6.9–8.2	5.4–6.8	4.0–4.9	3.0–3.9
Water absorption after 1 h	%	20.8	22.8	21.5	23.2

**Table 2 materials-16-06359-t002:** Compositions of Concrete with Different Fillers.

Name of the Concrete	Material Consumption Per 1 m^3^ (kg)				
	Cement, kg	Sand, kg	Filler		W/C (Water/Cement Ratio)
			5–10 mm	10–20 mm	
Ceramporite concrete type 1	330	539	335	503	0.76
Ceramporite concrete type 2	380	550	320	480	0.81
Heavy concrete	340	650	472	708	0.59

**Table 3 materials-16-06359-t003:** Characteristics of the reinforcing steel.

Reinforcing Steel Grade/Class	Diameter, mm, d	Yield Strength, MPa, σ0.2	Ultimate Strength, MPa, σu	Young’s Modulus, MPa E*_s_*·10^4^
K-7	9	1540	1765	191.0
K-7	12	1512	1777	193.8
Vr-II	5	1480	1694	212.3
Vr-I	5	-	574	170.0

**Table 4 materials-16-06359-t004:** Comparison of experimental and theoretical moments of crack formation.

Beam Code	$M_{crc}^{exp}$, kN⋅sm	$\frac{M_{crc}^{exp}}{M_{crc}^{exp}}$	Coefficient $K_{Rbt\cdot ser}$	Theoretical Values by Formula (1)		${∆}_{i}=\frac{M_{crc}^{exp}-M_{crc}^{i}}{M_{crc}^{exp}}$ × 100%	
				Excluding $K_{Rbt\cdot ser}$, kN⋅cm	Taking into account $K_{Rbt\cdot ser}$, kN⋅cm	${∆}_{1}$	${∆}_{2}$
1	2	3	4	5	6	7	8
BL19V-1	1595	0.46	0.91	1607.25	1566.0	−0.76	−1.82
BL19V-2	1567.5	0.457	0.91	1602.50	1563.0	−2.23	0
BL19S-1	1100	0.339	0.92	1070.95	1037.80	2.64	5.65
BL19S-2	1100	0.345	0.92	1076.87	1031.80	2.1	6.20
BL19O-1	330	0.104	1.0	352.70	-	−6.87	-
BL19O-2	335.5	0.103	1.0	352.70	-	−5.12	-
BL112B-1	2090	0.437	0.81	2239.30	2084.47	−7.14	0.26
BLP2B-2	2090	0.437	0.81	2184.25	2095.82	−4.51	0.28
BL112S-1	1925	0.4375	0.85	2095.54	2025.73	−8.86	−5.23
BLP2S-2	1980	0.439	0.85	2105.77	2035.96	−6.35	−2.82
BL112O-1	330	0.0774	1.0	328.60	-	0.4	-
BLP20-2	341	0.082	1.0	328.60	-	3.64	-
BLP9V-1	1320	0.444	0.9	1444.20	1403.43	−9.4	−6.32
BLP9V-2	1375	0.435	0.9	1438.90	1398.10	−4.64	−1.68
BLP9S-1	990	0.327	0.92	1069.37	1036.30	−8.02	−4.67
BLP9S-2	1100	0.37	0.92	1074.13	1041.06	−2.35	5.35
BLP9O-1	341	0.113	1.0	337.11	-	1.14	-
BLP9O-2	357.5	0.113	1.0	337.11	-	5.7	-
BLP12B-1	1925	0.4516	0.81	2101.43	2013.0	−9.16	−4.57
BLP12B-2	1925	0.446	0.81	2088.87	2000.44	−8.51	3.92
BLP12S-1	1787.5	0.442	0.85	1954.91	1887.02	−9.36	−5.56
BLP12S-2	1787.5	0.411	0.85	1979.38	1912.14	−10.73	6.97
BLP12O-1	302.5	0.0733	1.0	295.4	-	2.35	-
BLP12O-2	302.5	0.0733	1.0	295.40	-	2.35	-
BT9B-1	1210	0.478	0.82	1272.42	1215.50	−5.16	−0.45
BT9B-2	1210	0.42	0.82	1281.00	1224.18	−5.87	−1.17
BT9S-1	962.5	0.388	0.89	935.60	901.02	2.79	6.38
BT9S-2	962.5	0.388	0.89	940.10	905.52	2.33	5.92
BT9O-1	396	0.120	1.0	368.05	-	7.07	-
BT9O-2	385	0.117	1.0	368.05	-	4.41	-
BT12B-1	1815	0.455	0.73	1938.20	1839.10	−6.78	−1.32
BT12B-2	1787.5	0.471	0.73	1916.94	1817.77	−7.24	−1.69
BT12C-1	1787.5	0.433	0.73	1861.10	1716.03	−4.11	1.48
BT12C-2	1787.5	0.433	0.73	1896.50	1796.36	−6.1	−0.495
BT12O-1	319	0.077	1.0	299.40	-	5.95	-
BT12O-2	308	0.075	1.0	299.40	-	2.79	-

**Table 5 materials-16-06359-t005:** The influence of the upper load level and concrete type on the moment of crack closure during repeated unloading.

Beam Code	Upper Level Repeat. Load_ M/Mu	At First Unloading		At Fourth Unloading		$\frac{M_{3.4}}{M_{3.1}}$	Crack Opening Width under Loading		Coefficient
		Crack Closure Moment $M_{3.1}^{exp}$, kN · cm	Voltage Compression σ_bp_, MPa	Moment of Crack Closure $M_{3.4}^{exp}$, kN · cm	Compression Stresses σ_bp_, MPa		First ${ɑ}_{crc1}^{max},$ mm	Fifth ${ɑ}_{crc5}^{max},$ mm	$\varphi l\;\_=\frac{{ɑ}_{crc5}^{max}}{{ɑ}_{crc1}^{max}}$
1	2	3	4	5	6	7	8	9	10
BL19B-2	0.72	1100	0.73	1039.5	1.20	0.945	0.15	0.175	1.166
BL19C-2	0.6	797.5	−0.81	701.8	−0.08	0.88	0.15	0.17	1.133
BLP2B-2	0.78	1722.5	0.38	1650.0	0.905	0.96	0.15	0.16	1.07
BLP2S-2	0.76	1729.7	−0.75	1636.2	−0.1	0.95	0.15	0.16	1.07
BLP9B-1	0.79	1039.5	0.3	950.4	0.76	0.92	0.15	0.18	1.2
BLP9S-2	0.56	653.4	−0.06	594	0.4	0.91	0.15	0.165	1.1
BLP12B-2	0.81	1727	−0.66	1597.5	0.28	0.92	0.15	0.16	1.07
BLP12S-1	0.75	1439.9	0.33	1355.3	0.94	0.94	0.15	0.16	1.07
BT9B-2	0.66	953	0.7	895.1	1.16	0.94	0.15	0.17	1.133
BT9S-1	0.67	693	−0.5	643.5	−0.1	0.93	0.15	0.17	1.133
BT12B-2	0.84	1394.2	1.77	1251.3	2.85	0.90	0.15	0.17	1.133
BT12C-1	0.76	1395.6	1.29	1236.1	2.49	0.88	0.15	0.17	1.133

## Data Availability

Data is contained within the article.

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
