# Peer review of "Crack Resistance of Prestressed Reinforced Concrete Beams with Wire Rope Reinforcement"

_materials, 2023, doi:10.3390/ma16196359_

Round 1
Reviewer 1 Report
The crack resistance of prestressed lightweight reinforced concrete flexural elements with wire rope reinforcement was studied. The following comments should be addressed before further consideration.
(1) Line 34: The reference number starts with 5, not 1?
(2) The roadmap of the test design can be drawn in Section 2.
(3) Some relative test figures of the materials and test machine can be added.
(4) The crack and damage was conducted in this paper. Generally, the strength is closely related with the damage characteristics, which could be analysed with the analytical model (Ref to https://doi.org/10.1142/S1758825123500369). The results of shear damage analysis can provide a fundamental basis for stability analysis in geotechnical engineering. The shear damage model and characteristics can be reviewed and added in the introduction or relative section.
(5) The variables in Figure 1 should be illustrated. Is the horizontal axis the strain? Then the unit is not correct. Should it be 10^(-5)?
(6) It is better to add the figures showing the progressive failure in the test.
(7) More recent studies of the progressive failure in the explanation of the damages in structures can be reviewed, such as Zhen Zhang, Zhen Li, Gang Xu, et al., Lateral abutment pressure distribution and evolution in wide pillars under the first mining effect, International Journal of Mining Science and Technology, 33(3): 309-322.
(8) The units in the formula should be added.
(9) Uncertainty analysis is an important challenge to stability analysis (A practical and efficient reliability-based design optimization method for rock tunnel support). How to use the results with optimum parameters for engineering could be discussed.
(10) The inspiration for the application in future can be discussed.
Author Response
The response file is attached.

Reviewer 2 Report
1) In the Introduction, some references dealing with reinforced concrete structures should be mentioned, as:
Frappa G, Pauletta M. Seismic retrofitting of a reinforced concrete building with strongly different stiffness in the main directions. Proceedings of 14th fib International PhD Symposium in Civil Engineering, 5-7 September 2022, Rome, Italy.
2) At the end of the Introduction, the research golas should be ezplained in a more detailed and clear way.
3) In Section 2, some references dealing with lightweight and heavyweight concrete should be mentioned. With regard to heavyweight concrete the following work could be cited:
Frappa G, Miceli M, Pauletta M. Destructive and non-destructive tests on columns and cube specimens made with the same concrete mix. Construction and Building Materials 349 (2022) 128807.
4) In Section 2 some images of the test specimens should be provided. Also test setup, including load application and restraints applied to the beams, should be presented. Moreover, it is important to add figures showing the cross sections of tested beams and the layout of the longitudinal and transverse reinforcements. In addition mechanical properties of the the materials (concrete and reinforcements) must be specified.
5) In Section 3, some images showing the evolution of the crack pattern as the load increases should be presented. Moreover, the meaning of all terms appearing in the equations should be specified. Also the expression of Kbtser must be provided. In addition, the labels of the cracking moments appearing in Table 1 are different from those specified in the text.
There are some typos spread throughout the paper.
Moreover, the last sentence in line 216 must be checked.
Author Response
The response file is attached.

Reviewer 3 Report
Good research paper. Reviewer likes the topic investigated in the journal paper submitted. The investigation is clear in all its parts, well written and well balanced. As such, this reviewer has no hesitations in recommending its acceptance for publication in Materials.
Nonetheless, this reviewer would like to bring a few minor comments/suggestions to the attention of the authors, which will hopefully increase the quality of the manuscript even further.
- Abstract is very short. Just a few general information are described. Please add a couple of sentence more to go deeper into the research field.
- May the authors clarify what is the meaning of “Bendable elements made of reinforced cable-reinforced ceramsite concrete”. It seems an uncommon terminology. Particularly “bendable” and “ceramsite”
- Authors say “it is necessary to carry out new experimental and theoretical research to refine and improve the calculation and design requirements of the regulatory documents for the design of such prestressed reinforced concrete structures.” This is partially true but please refer in the bibliography to the following very recent researches that can be considered seminal: DOI 10.1186/s40069-023-00580-w, DOI 10.5459/bnzsee.1572
- May the authors be more clear in the “Introduction” related to “peculiarities of prestressed structures”. Which kind of peculiarities. Again the “Introduction” as the Abstract is very short.
- May the authors add some figures (photos) of the 36 experimental reinforced concrete beams that were fabricated
Author Response
The response file is attached.

Reviewer 4 Report
Dear Authors,
This study presented the results of Crack resistance of prestressed reinforced concrete beams with wire rope reinforcement. Manuscript is not prepared on the good way and it have to be improved according to instruction. Recommendations includes:
1. Expand the summary to include the important results.
2. Rewrite the introduction, emphasising the material studied in your work.
3. What is new in your manuscript that needs to be visible after the introduction?
4. Rewrite the experimental section because it is full of errors regarding the methods and materials used.
5. Nowhere is it stated how the samples were prepared.
6. The measurement methods are not sufficiently described!
7. Add more references to previously published work in the "Results and Conclusions" chapter!
8. Significantly increase the number of references to the research topic.
Dear Sir/Madam,
The moderate editing of English language is required
Author Response
The response file is attached.

Round 2
Reviewer 1 Report
I have come through the revised version and have no more queries. It could be considered to be accepted.
Author Response
The file is attached.

Reviewer 2 Report
1) In the Introduction, some references dealing with reinforced concrete structures should be mentioned, as:
Frappa G, Pauletta M. Seismic retrofitting of a reinforced concrete building with strongly different stiffness in the main directions. Proceedings of 14th fib International PhD Symposium in Civil Engineering, 5-7 September 2022, Rome, Italy.
2) In Section 2, some references dealing with lightweight and heavyweight concrete should be mentioned. With regard to heavyweight concrete the following work could be cited:
Frappa G, Miceli M, Pauletta M. Destructive and non-destructive tests on columns and cube specimens made with the same concrete mix. Construction and Building Materials 349 (2022) 128807.
Language is quite good.
Author Response
The file is attached.

Reviewer 3 Report
Authors properly answered all the reviewer's comments.
Author Response
The file is attached.

Reviewer 4 Report
Dear Authors,
In the introduction a large number of references are inserted in only one sentence "The studies by Dmitriev S.A., Gvozdev A.A. [5, 6, 7], Mikhailov V.V., Mikhailov K.V. [8, 9], Berdichevsky G.I. [10], Gushchi Yu.P., Semenov A.I. [11, 12, 13], Mailian R.L. [14,15], Askarov B.A. [16, 17, 18], Semenov A.I. [19], and Hakimov Sh.A. [20] are dedicated to the investigation of the peculiarities of prestressed structures.". This is just a way to increase the number of references, but it is not appropriate. Please use them when reformulating the introduction.
Line 28/29:"... filler obtained by firing a mixture of clay, kaolin, and bentonite clays)" What kind of clay, bentonite is also clay? Remove parenthesis from the sentence.
Line 33: "Bendable (capable of being..." Remove parenthesis.
In Table 1, a period (.) must be used instead of a comma (,) at the decimal point
In the Materisla and Methodes:
Please provide more information about the ceramisite used (how it is produced, under what conditions, particle size distribution, and mineralogical or chemical properties! This must be part of the manuscript.
line 96/97: What type of cement do you use? Provide some information on the chemical, mineralogical and physical properties of cement.
Is there any information missing from the manuscript on the production of concrete?
In the chapter Results and Disscusion, the recommendation to increase the number of references is not accepted.
Moderate editing of English language required
Author Response
The file is attached.
